# Development and Validation of a 54K Genome-Wide Liquid SNP Chip Panel by Target Sequencing for Dairy Goat

**DOI:** 10.3390/genes14051122

**Published:** 2023-05-22

**Authors:** Shengyu Guan, Weining Li, Hai Jin, Lu Zhang, Guoshi Liu

**Affiliations:** 1State Key Laboratory of Animal Biotech Breeding, Key Laboratory of Animal Genetics and Breeding of the Ministry of Agriculture, Beijing Key Laboratory for Animal Genetic Improvement, College of Animal Science and Technology, China Agricultural University, Beijing 100193, China; gsy729@hotmail.com (S.G.); liwn@cau.edu.cn (W.L.); luzhang2018@cau.edu.cn (L.Z.); 2Inner Mongolia Academy of Agricultural & Animal Husbandry Sciences, Hohhot 010031, China; jinhaicnm@vip.sina.com

**Keywords:** SNP chip, genotyping by target sequencing, GBTS, dairy goat breeding, sequence capture in-solution (liquid chip)

## Abstract

As an important genotyping platform, SNP chips are essential for implementing genomic selection. In this article, we introduced the development of a liquid SNP chip panel for dairy goats. This panel contains 54,188 SNPs based on genotyping by targeted sequencing (GBTS) technology. The source of SNPs in the panel were from the whole-genome resequencing of 110 dairy goats from three European and two Chinese indigenous dairy goat breeds. The performance of this liquid SNP chip panel was evaluated by genotyping 200 additional goats. Fifteen of them were randomly selected for whole-genome resequencing. The average capture ratio of the panel design loci was 98.41%, and the genotype concordance with resequencing reached 98.02%. We further used this chip panel to conduct genome-wide association studies (GWAS) to detect genetic loci that affect coat color in dairy goats. A single significant association signal for hair color was found on chromosome 8 at 31.52–35.02 Mb. The TYRP1 gene, which is associated with coat color in goats, was identified to be located at this genomic region (chromosome 8: 31,500,048-31,519,064). The emergence of high-precision and low-cost liquid microarrays will improve the analysis of genomics and breeding efficiency of dairy goats.

## 1. Introduction

Applications of molecular-assisted markers have greatly improved the efficiency and accuracy of livestock selection and breeding [1]. The transition from “empirical breeding” to “precise breeding” with high efficiency not only reduces the shortcoming of conventional breeding, such as the uncertainty of the trait selection, but also significantly improves breeding efficiency and accuracy to develop the new high-yielding and high-quality breeds. Genomic selection (GS) is one of the landmark technologies of molecular breeding. It is characterized by the ability to use genome-wide genetic markers of animals, not just their phenotypes, to guide livestock selection and breeding. Since GS propulsion in 2001 [2], it has revolutionized the field of animal breeding. In the past two decades, genomic selection methods have been successfully applied to breeding practices in many livestock, including cattle [3,4,5], pigs [6,7,8], and chickens [9,10], with satisfactory results.

The accuracy of GS relies on SNP loci covering the whole genome; therefore, the high-throughput variant locus detection technology has been one of bottlenecks in GS. The development of the SNP chip array has advanced the application of GS. More recently, a liquid SNP chip panel based on genotyping by targeted sequencing (GBTS) technology has been developed to reduce the cost further and improve the accuracy of genomic selection. Compared with traditional SNP chips using magnetic beads, liquid-phase chips have many advantages, including low cost and high flexibility [11,12]. It has also been applied to the development of SNP chip arrays for many plant and animal species. For example, the recently developed 40K SNP liquid chip for maize has reduced the genotyping cost to as low as $14 [13].

Dairy goats are widely bred worldwide. Traditional breeding methods for dairy goats rely on manual experience and observation, which have low efficiency, long cycle time, and susceptibility to environmental factors. In contrast, genetic selection is a well-established method for a more objective and accurate assessment of production performance. However, this method has not yet been utilized in Chinese dairy goat breeding due to the small number of phenotype records involved and the lack of commercial SNP chips. Although GoatSNP50 Bead Chip [14] from Illumina Inc. is widely used in GWAS studies for dairy goats [15], new SNP chips with low cost and easy application are needed for Chinese dairy goat genome selection. 

This article presents the development and validation of a 54K genome-wide liquid SNP chip panel for targeting dairy goat sequencing. The chip contains SNP-associated production traits from genomic regions. The chip was validated by genotyping 200 dairy goats and comparing their liquid SNP chip panel and WGS data. Developing a 54K genome-wide liquid SNP chip panel using target sequencing is a significant step forward in dairy goat breeding.

## 2. Materials and Methods

We are committed to ensuring that the ethical principles of animal experimentation are fully observed to ensure that the welfare and rights of animals are fully protected. All experiments were performed under the approval of the Ethics Review Committee of China Agricultural University (permit number: AW12401202-1-32).

### 2.1. Biological Material, DNA Extraction, and Sequencing

The resequenced animals used in this study consisted of five dairy goat breeds, including three European dairy goat breeds, namely Saanen dairy goats (*N* = 25), Toggenburg dairy goats (*N* = 27), and Alpine dairy goats (*N* = 16), and two Chinese indigenous dairy goat breeds, Guanzhong dairy goats (*N* = 19) and Laoshan dairy goats (*N* = 23). Genomic DNA from these 110 goats was extracted from ear punches (preserved in ethanol, -20 degrees Celsius) using Qiagen DNeasy columns (Qiagen, Hilden, Germany) following the manufacturer’s instructions. The extracted DNA was quantified using a Qubit fluorometer, and the quality was assessed using a Nanodrop spectrophotometer. Sequencing libraries were prepared and barcoded using the TruSeq sample preparation kit. The libraries were sequenced on the Illumina HiSeq X Ten platform using the 150 bp paired-end sequencing strategy.

### 2.2. Trimming, Mapping, and SNP Detection

All WGS reads were checked for sequencing quality using FastQC [16] and trimmed using TrimGalore (version 0.6.5) [17] according to the following parameters: “-g 20 -phred33—stringency 3—length 20-e 0.1.” All cleaned reads were mapped to the goat reference assembly ARS1 (GCF_001704415.1) using BWA-MEM (version 0.7.17) [18] with the default parameter.

To obtain high-quality SNPs for chip development, a Genome Analysis Toolkit (GATK, version 4.2.0.0) [19,20,21] was used to extract the variant according to the following parameters: “QD < 2.0,” “QUAL < 30.0,” “SOR > 3.0,” “MQ < 40.0,” “FS > 60.0,” “MORankSum < −12.5,” and “Read PosRankSum < −8.0.” 

After this, PLINK1.9 [22] was used to further extract variants of 29 autosomal chromosomes and 2 contigs, which are unlocalized-scaffold of chromosome X. The filter settings were as follows: SNPs with missing rates (Miss) ≥0.1, and minor allele frequencies (MAF) < 0.05. Finally, a total of 33,716,044 SNPs remained for further analyses.

### 2.3. Development of the Dairy Goat SNP Panel

To begin with, we conducted selective sweeps according to differences between Chinese and European dairy goat breeds in milk production, milk quality, and other production traits. Selective signal sweep regions were detected according to combinations of three parameters: the XP-CLR approach implemented in the XP-CLR (version 1.1.2) [23], fixation index value (Fst), and nucleotide diversity (π) estimated using vcftools (version 0.1.16). Secondly, QTL loci related to fore udder (FU), feet, and legs (FL), general appearance (GA), rear udder (RU), suspensory ligament (SL), and teats (TE) were obtained from the goat QTLdb database [24] and overlapped with SNPs in the resequencing dataset to get candidate loci. Thirdly, we performed a GWAS analysis based on collected phenotypic information of the 110 resequenced individuals. Association analyses of 305 days of milk yield (MY), milk protein content, somatic cell count, and melatonin synthesis capacity were performed using the MLM in the GEMMA/0.96 [25]. Finally, to ensure uniform SNP density, we filled gaps using loci from the resequencing dataset with a MAF >0.5, and 62,492 candidate SNPs were selected. 

After scoring according to the Targeted Capture Sequencing Probe Design System (Compass Biotechnology, Guilin, China), the final 54,188 loci were obtained. A total of 53,990 probes were designed based on the 54K loci, and all probes were 120 bp in length.

### 2.4. Validation of the 54K Liquid SNP Chip Panel

To get a more comprehensive overview of the panel and SNP performance, 200 additional dairy goat samples were genotyped using the 54K liquid SNP chip panel, of which 15 individuals were randomly selected for 30× whole-genome resequencing. The sequencing process and analysis of the raw data for the liquid SNP chip panel and resequencing are the same as in Section 2.1 and Section 2.2 above.

### 2.5. Genome-Wide Association Study

To eliminate the contribution of information from relatives, the de-regressive estimated breeding value (DEBV) calculated according to the method proposed by Garrick et al. [26] was used as the response variable in the GWAS in this study. The GWAS was performed with the following single-marker regression model by the GEMMA Software:y=Xm+Wa+e
where *y* is the vector of the response variable (DEBVs in this study), *m* is the vector of SNP marker effects, and *a* is the vector of residual polygenic effects with a normal distribution $a∼N\left(0,G\sigma_{a}^{2}\right)$, *G* is the realized relationship matrix constructed with markers and $\sigma_{a}^{2}$ is the additive genetic variance, *e* is the vector of residual errors with a normal distribution $e∼N\left(0,I\sigma_{a}^{2}\right)$, *I* being an identity matrix and $\sigma_{e}^{2}$ the residual variance, and *X* and *W* are incidence matrices of *y* related to m and *a*, respectively.

## 3. Results

### 3.1. Design and Development of the 54K Liquid SNP Chip Panel

The sequencing of 110 dairy goats generated 72,083.59M reads, representing an average of 32.45 × coverage per individual (min = 25.3, max = 39.2). After alignment, on average, 99.84% of reads were mapped to the ARS1 assembly. In total, 33,716,044 SNPs were eventually obtained for subsequent chip development. After a series of filtering steps in the material approach (Figure 1A), the final 54,188 loci were identified. 54K loci are evenly distributed in the genome and cover the two contigs LWLT01000021.1 LWLT01000027.1, which were used to represent the unassembled X chromosome of the ARS1 reference genome (Figure 1B). Most of the SNPs are spaced within 100 KB (Figure 1C).

### 3.2. Genotyping Performance of the Liquid SNP Chip Panel

To validate the genotyping performance of the liquid SNP chip panel, 200 dairy goats were genotyped. The average number of reads obtained from targeted capture sequencing reached 14,927,888, of which the percentage of reads localized to the reference genome was 99.97% on average (Appendix A Appendix A). The average number of double-ended reads sequenced to the reference genome and at a distance that matched the length distribution of sequenced fragments (properly mapped ratio) reached 95.61% of all reads, with a small number of individuals below 92% (Figure 2A).

The detection efficiency of the 54K target loci was judged according to the sequencing depth of the loci. The average detection rate was 98.82% for sequencing depths greater than 0 (dp0%), the average detection efficiency decreased to 98.41% for sequencing depths greater than 5 (dp5%), and the average detection efficiency was 97.22% for sequencing depths of 20 (dp20%) as the threshold (Figure 2C, Appendix A Appendix A). Analysis of MAF values for captured SNPs showed that 97.04% had MAF values >0.05 (Figure 2B).

As a characteristic of GBTS, the flanking region of the target locus has a high sequencing depth and can capture variants beyond the designed target locus. In this experiment, a total of 100K high-quality SNPs were detected, which was twice as many as the target loci. Meanwhile, the results showed that the average sequencing depth of the target loci reached 91X, and the two flanking loci still had 60X and 26X sequencing depths at distances of 100 and 250 bp, respectively (Figure 2D, Appendix A Appendix A).

### 3.3. Comparison of Chip Panel and Resequencing Results

To investigate the accuracy of the liquid SNP chip panel detection effect, we randomly selected 15 individuals for simultaneous liquid SNP chip panel detection and whole-genome resequencing at a sequencing depth of 30X. It was shown that 98.02% (Figure 3A) and 97.78% (Figure 3C) of 54K target SNPs and a total of 100K SNP are the same as the WGS result. After further analyzing the loci with inconsistent genotypes, the causes of inconsistent loci were classified as follows: “GBTS miss” as no SNP was detected by GBTS, while SNP variants were detected in WGS (GBTS miss), on the opposite “WGS miss.” “GBTS hom” as only one allele was detected on GBTS (monomorphic genotype), while two alleles were detected in WGS (polymorphic genotype), on the opposite “WGS hom.” It was found that among the 54K SNP loci, 96.73% of the loci generated errors due to GBTS (GBTS miss+ GBTS hom) (Figure 3B). Among the total 100K SNP loci, the errors due to GBTS were 85% (GBTS miss+ GBTS hom) (Figure 3D). Among the 54K SNP loci, the “GBTS miss” was over 50%, while among the 100K SNP loci, “GBTS hom” dominated, accounting for 62.58%.

### 3.4. High-Resolution GWAS for Coat Color

To evaluate the effect of the liquid SNP chip panel, we conducted GWAS to detect genetic loci that affect coat color in dairy goats. In our sequenced population, the Toggenburg dairy goat had a signature brown pattern, while the Saanen dairy goat was pure white. We used the 54K SNP and 100K SNP datasets for detection (Figure 4). GWAS for this trait detected 7 and 23 associated SNPs with genome-wide significance. A significant association signal was found on chromosome 8 at 31.52–35.02 Mb. The TYRP1 gene is located in this genomic region (chromosome 8: 31,500,048–31,519,064).

## 4. Discussion

Using reliable and low-cost genotyping strategies is essential in animal breeding, including dairy goats. In this study, using resequencing data from Chinese and European dairy goat breeds, we developed a dairy goat liquid SNP chip panel. The main purpose is to set the scope using this chip, mainly for Chinese dairy goat breeding. In the process of SNP selection, trait relatedness was considered an important indicator. The specific traits covered by the liquid SNP chip panel included milk yield, fat yield, protein yield, fore udder, feet and legs, general appearance, rear udder, suspensory ligament, and teats. In addition, SNPs associated with melatonin synthesis capacity were also included. This is because in our previous work, melatonin synthesis capacity was found to play a vital role in milk-producing livestock and can be a critical selection indicator [27,28,29].

This 54K genome-wide liquid SNP chip panel is based on GBTS. After testing and verification, the designed capture efficiency of 54K SNPs can reach 98.41%, and the consistency of capture loci and resequencing can also reach 98.02%. Among other breeding chips based on targeted capture sequencing, the capture efficiency of the current chip can be considered highly satisfactory [30,31,32]. In addition, the GBTS could obtain variant SNP other than the target capture SNP. In previous publications, this was referred to as the “multiple single-nucleotide polymorphism (mSNP) approach” [33,34]. We also captured over twice as many high-quality SNP variants at the designed loci in the liquid SNP chip panel. Compared to other reports, we detected relatively fewer SNPs. This may be related to the fact that the conditions for screening consistent with resequencing were different in the variant screening process and/or the variable probe length, as well as in the size of our chip detective population. When comparing the resequencing results with the multi-capture SNP, it was found that more than 97% of the SNP were identical, further demonstrating the high quality of mSNPs obtained by targeted capture sequencing.

By performing GWAS analysis on dairy goat hair color using the captured 54K SNP dataset and 100K SNP dataset, respectively, TYRP1, a gene associated with goat and other animals’ coat color [35,36,37,38], was successfully localized at chromosome 8: 31,500,048-31,519,064. Meanwhile, it was found that the Manhattan and QQ plots produced by the 100K dataset were better than the 54K SNP dataset. Due to the difficulty of recording phenotypic data of the experimental population, it is not practical to perform GWAS studies for other traits. However, it is possible to find more associated SNPs if more individual phenotypes can be obtained, especially with twice as many SNPs as target loci. By analyzing the reliability of GBTS for SNP detection, the data showed that the full range of SNP obtained using GBTS is equivalent or even superior to the target SNP. This finding further illustrates the advantages of liquid-phase microarrays based on targeted capture sequencing.

Of course, the liquid SNP chip panel-based GBTS had some shortcomings during the analysis. By comparing liquid SNP chip panel detection to resequencing of genotypically inconsistent SNP, it was found that most errors came from GBTS, where variants were not detected, or heterozygous alleles were considered to be homozygote alleles. The reason for this phenomenon is most likely related to the allele drop-out (ADO) that occurs during the PCR process. This occurs when only one of the two alleles present in a sample is amplified to a detectable level [39]. This phenomenon indicates that GBTS is not as accurate as resequencing. 

Compared with SNP chips based on the magnetic bead principle, liquid SNP chip panels based on GBTS have obvious advantages, including low cost and a higher number of detection SNPs than design SNPs. In addition, the quantity of liquid SNP chip panels can be produced according to the required testing quantity, which further reduces costs. However, due to the longer time for data analysis and processing with liquid SNP chip panels, this is considered a disadvantage compared with the bead SNP chip. This disadvantage may have its benefit, because as the detection population expands, the SNPs that are not originally detected as polymorphic may be re-identified, resulting in a richer set of variant SNP datasets.

## 5. Conclusions

Overall, a low-cost, high-accuracy dairy goat whole-genome liquid chip based on resequencing data was developed. This chip can be widely used in GWAS and GS for large-scale genotyping, offering a new tool for the molecular breeding and genetic improvement of dairy goats.

## Figures and Tables

**Figure 1 genes-14-01122-f001:**
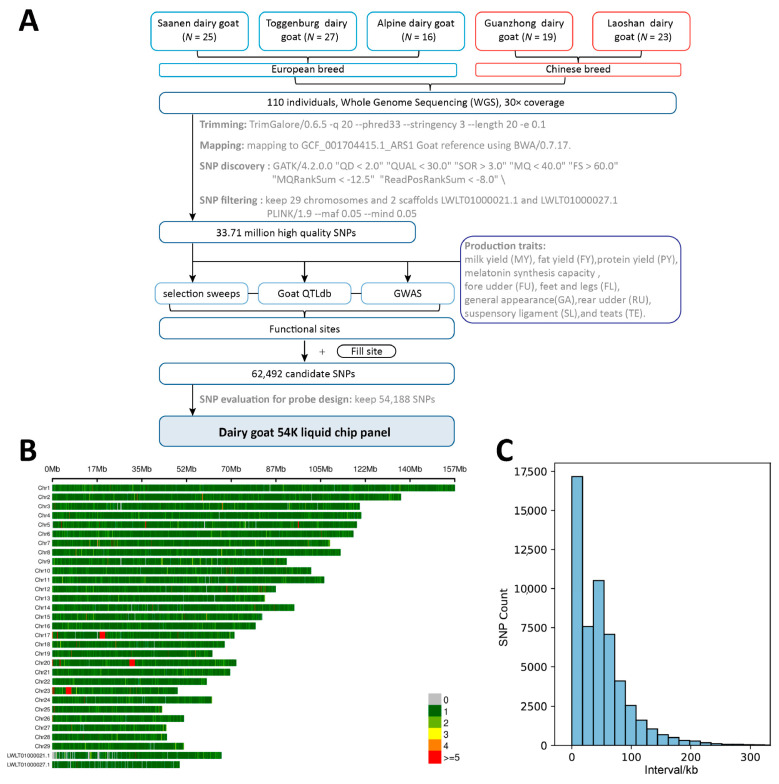
Design and development of the 54K liquid SNP chip panel. (**A**) The graphical summary of the pipeline for liquid SNP chip panel design. (**B**) 54K SNP density of liquid SNP chip panel. (**C**) Spacing of adjacent SNPs.

**Figure 2 genes-14-01122-f002:**
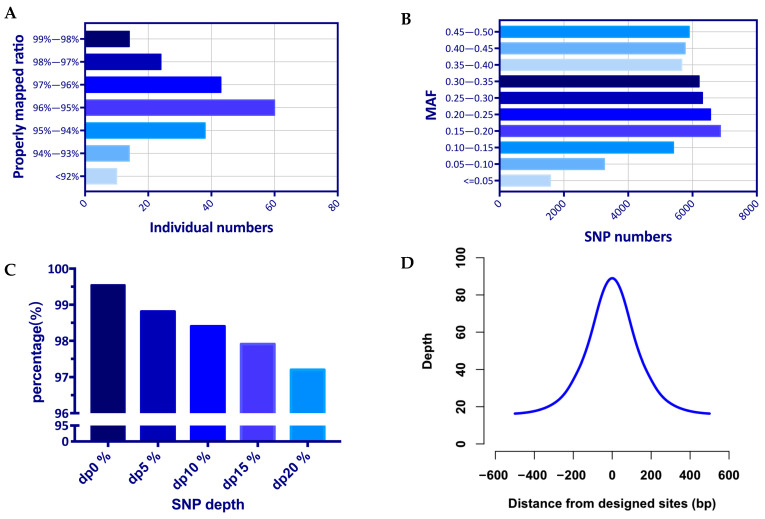
Genotyping performance of the liquid SNP chip panel: (**A**) reads properly mapped ratio; (**B**) minor allele frequency (MAF) of captured 54K SNPs; (**C**) average detection rate of different sequencing depths; (**D**) average sequencing depth of the target loci and 500 bp flanking region.

**Figure 3 genes-14-01122-f003:**
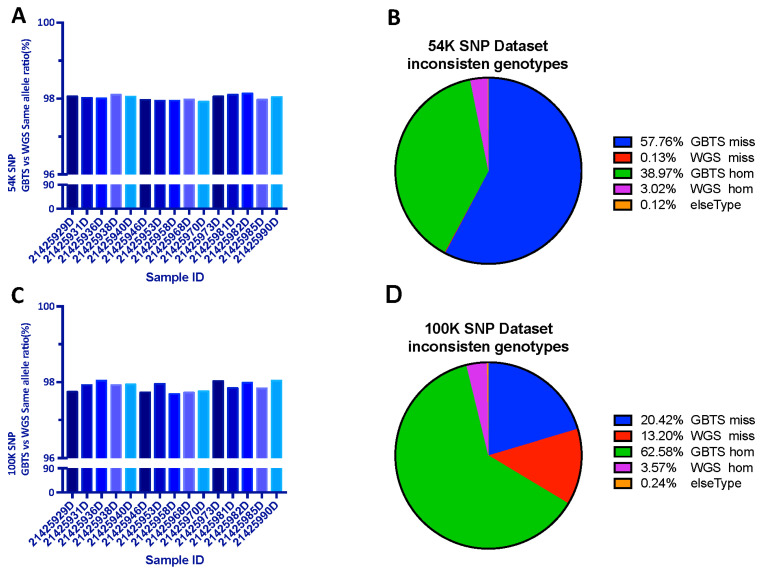
Comparison of chip panel and resequencing results: (**A**) 54K SNP dataset with same allele ratio; (**B**) classification of inconsistent causes in the 54K dataset; (**C**) 100K SNP dataset with same allele ratio; (**D**) classification of inconsistent causes in the 100K dataset.

**Figure 4 genes-14-01122-f004:**
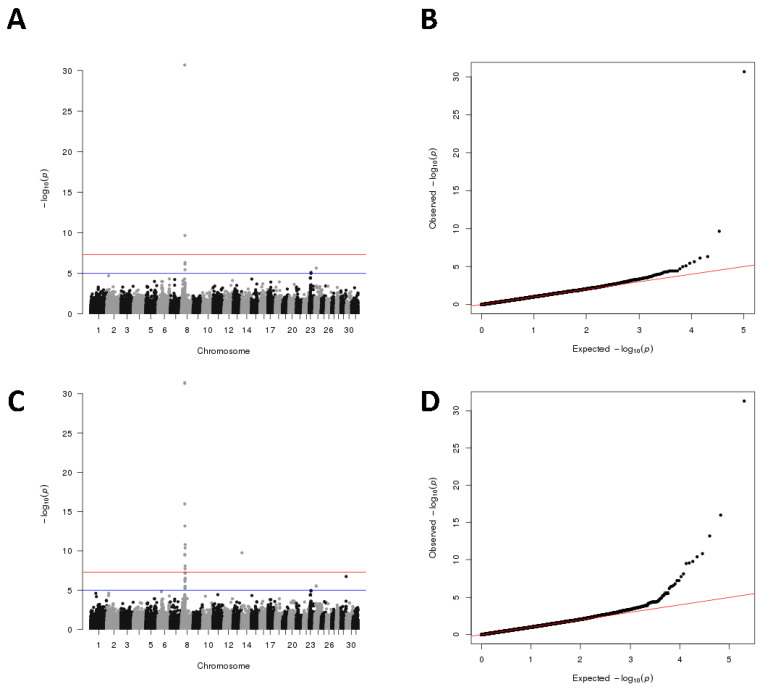
High-resolution GWAS for coat color: (**A**,**B**) Manhattan and quantile-quantile plot of GWAS for coat color using 54K SNP dataset. The red line indicates the threshold for genome-wide significance (*p* < 5 × 10^−8^) and the blue line for suggestive associations (*p* < 1 × 10^−5^); (**C**,**D**) Manhattan and quantile-quantile plot of GWAS for coat color using 100 K SNP dataset.

## Data Availability

The data presented in this study are available on request from the corresponding author. The data are not publicly available due to the experiment based on these data is not over yet.

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
