# Peer review of "Development and Validation of a 54K Genome-Wide Liquid SNP Chip Panel by Target Sequencing for Dairy Goat"

_genes, 2023, doi:10.3390/genes14051122_

Round 1

Reviewer 1 Report

The authors have generated and validated a SNP panel for Chinese Dairy Goat breeding that utilizes a genotype by targeted sequencing approach.  The design is sufficient, and sound, with the results being supported by the data as shown.  The authors have generated a new tool for dairy goat molecular breeding that should be an advance of importance.

I have a few minor comments for the authors to consider.

Line 29  "In the era of breeding, 3.0" ?? Not sure this is a common or very descriptive term for what you are trying to say.

Lines 42-45  Are there any more recent references for the liquid SNP chip panel approach rather than the one you reference from 2010??

Line 57  " can welly do" ??

Lines 72-75  Suggest splitting into two sentences.  Difficult to follow.

Line 98  "European breeds goat breed" ???

Line 249  for example describe how much less than other literature reports?

Line 270  I assume these errors are associated with allele drop-out when the target amplification primers encounter another previously undetected SNP??

Line 278  delete "that was"

There are relatively minor issues with the quality of the English Language.

Reviewer 2 Report

Guan et al. presented the development of a 54K SNP panel for genotyping dairy goats. Since the authors submitted the manuscript as a communication paper, I believe the content is suitable. However, I have several concerns about the experience design and validation.

1.       Line 99-102; The authors performed the signature of selections, but the methods and threshold are not clearly mentioned. The authors might add details in the supplementary files.

2.       Line 102-104: Why did the authors select the QTL for these traits, how are QTL for general production traits and health/disease-related traits?

3.       Line 105-107: The authors performed GWAS using 110 animals, I do not think it makes sense.

4.       Line 111-113: What is a threshold for scoring SNPs, and how is the score reported, more information is needed here.

5.       Line 124:  How DEBVs were calculated, and which methods were used for the calculation of DEBVs?

6.       How many SNPs overlap with the 52K SNP chip for goats?  The authors should discuss the cost and implementation of this panel compared to the 52K chip.

7.       For the reproductivity of the results, the generated data should be deposited?

8.       It is not clear about the comparison with 100 K.

9.       Figure 1A. Are all these animals sequenced in this project or are some from the publicly available data?

10.   The authors should proofread the manuscript carefully.

Line 15: What did the authors mean by SNP performance?

Line 84, 88, 98-101, etc.: Add citation for the software used,

Line 214: define an abbreviation for DEBVs

Line 226: What did the authors mean by “high resolution”?

The authors should proofread the manuscript carefully. The current version of the manuscript is not ready for publication. 

Round 2

Reviewer 2 Report

The authors have addressed my comments, some minor suggestions:

Line 103: The authors do not need two references for Plink, it should be the one for the version used.

Line 106: Might change were obtained to were remained for further analyses.

Line 135:  The authors should add sentences about how DeEBVs calculated in the main manuscript.

The results of GWAS is quite surprising, the authors got very high significant signals given the data are small. Did the authors check the distribution of DeEBVs

Line 213: In figure 2, define MAF as minor allele frequency.

Line 251-254; Some abbreviations might not need here. 

Line 262: Defining abbreviation (GBTS) is not necessary here.

Some minor checks are required 
